# Ultrasound Methods for the Assessment of Liver Steatosis: A Critical Appraisal

**DOI:** 10.3390/diagnostics12102287

**Published:** 2022-09-22

**Authors:** Dorotea Bozic, Kristian Podrug, Ivana Mikolasevic, Ivica Grgurevic

**Affiliations:** 1Department of Gastroenterology and Hepatology, University Hospital Center Split, Spinčićeva 1, 21 000 Split, Croatia; 2Department of Gastroenterology and Hepatology, University Hospital Center Rijeka, Krešimirova 42, 51 000 Rijeka, Croatia; 3Department of Gastroenterology, Hepatology and Clinical Nutrition, University Hospital Dubrava, Avenija Gojka Šuška 6, 10 000 Zagreb, Croatia; 4School of Medicine, University of Zagreb, Šalata 2, 10 000 Zagreb, Croatia

**Keywords:** liver steatosis, steatosis quantification, NAFLD, ultrasound attenuation imaging, ATI

## Abstract

The prevalence of the non-alcoholic fatty liver disease has reached major proportions, being estimated to affect one-quarter of the global population. The reference techniques, which include liver biopsy and the magnetic resonance imaging proton density fat fraction, have objective practical and financial limitations to their routine use in the detection and quantification of liver steatosis. Therefore, there has been a rising necessity for the development of new inexpensive, widely applicable and reliable non-invasive diagnostic tools. The controlled attenuation parameter has been considered the point-of-care technique for the assessment of liver steatosis for a long period of time. Recently, many ultrasound (US) system manufacturers have developed proprietary software solutions for the quantification of liver steatosis. Some of these methods have already been extensively tested with very good performance results reported, while others are still under evaluation. This manuscript reviews the currently available US-based methods for diagnosing and grading liver steatosis, including their classification and performance results, with an appraisal of the importance of this armamentarium in daily clinical practice.

## 1. Introduction

The prevalence of the non-alcoholic fatty liver disease (NAFLD) has reached major proportions, with it estimated to affect one-quarter of the global population, and is showing a progressive trend into the future [1]. Potentiated by the global pandemic of obesity, along with metabolic syndrome, it might lead to non-alcoholic steatohepatitis (NASH), liver fibrosis, cirrhosis and the development of hepatocellular carcinoma (HCC). The burden of NAFLD is, therefore, tremendous, and the prevention of the disease and the early recognition of its complications are of the crucial importance.

B-mode ultrasound (US) has been the most commonly used method in the assessment of hepatic steatosis (HS) for a long time. However, its weak points have been revealed over the years, including the high inter-observer variability, low sensitivity for detection of mild steatosis (liver fat content <20%) and lower accuracy in patients with liver fibrosis [2]. 

According to World Federation for Ultrasound in Medicine and Biology (WFUMB), two methods are considered reference standards in diagnosing HS: liver biopsy (LB) and the magnetic resonance imaging proton density fat fraction (MRI-PDFF) [3]. Depending on the percentage of hepatocytes with fatty infiltration, a pathohistological grading system for HS has been developed, and it serves as the reference for testing the accuracy of developing non-invasive methods (Table 1) [4]. However, LB also has its disadvantages, such as its invasiveness, small sample volume and patient discomfort. 

MRI-PDFF on the other hand is a highly accurate, non-invasive and reproducible method. MRI- PDFF is expressed as a percentage and defined as the fraction of the signal from moving protons in triglycerides divided by the total proton signal from all mobile proton species [5]. Using histology as the standard, a cut-off of ≥5% of the fat fraction has been suggested as diagnostic for HS [5]. MRI-PDFF correlates well with the histologically proven steatosis grades and is considered superior to other non-invasive techniques in the quantification of the liver fat content [6]. It is, therefore, considered by many experts in field the best non-invasive tool that captures the histologic, pathophysiologic and metabolic features of the disease [7]. Unfortunately the high cost, limited availability and patient discomfort (e.g., claustrophobia) preclude its use in daily clinical practice. 

Hence, there has been a strong need for the development of new inexpensive and non-invasive techniques. The most appropriate and least demanding for daily use, especially among general practitioners, are biochemical scores, including the fatty liver index (FLI), hepatic steatosis index (HSI), Dallas steatosis index (DSI), NAFLD liver fat score and Framingham steatosis index (FSI) [8,9,10,11]. 

Both the FLI and HSI include components that are routinely performed (waist circumference, BMI and laboratory values) or information that is easily obtained (the presence of diabetes mellitus), and have been proven to have very good accuracy (AUROCs of 0.83 and 0.81, respectively) in comparison with the LB for the diagnosis of steatosis (≥S1) in the cohort of 324 patients with suspected NAFLD [8,12]. The recently developed DSI is based on magnetic resonance spectroscopy (MRS) data and includes information regarding the patient’s age, sex, ethnicity, laboratory values, BMI and comorbidities, but lacks external validation [9]. 

Several scores assess the risk of fibrosis development in patients with NAFLD, including the NAFLD fibrosis score (NFS), FIB-4, as well as the BARD and APRI scores [13]. They are used to exclude patients with advanced fibrosis, and possibly to identify ones at high risk for liver-related morbidity and mortality, whereas the NFS and FIB-4 outperform the APRI in mortality stratification [13,14]. Studies suggest that the aforementioned scores can be used for the risk stratification of NAFLD patients in clinical practice, but have limited performance in predicting fibrosis changes [13].

However, biochemical scores have not been implemented in clinical practice as was predicted [8]. The anamnestic data, clinical status, laboratory results and ultrasound findings seem to be enough for a diagnosis of NAFLD, without the requisite for scores that only confirm the presence of but do not grade HS. The further need to accurately quantify HS led to the development of the controlled attenuation parameter (CAP), a technique that estimates the hepatic fat content during the liver stiffness measurement (LSM) using vibration-controlled transient elastography (VCTE) via the FibroScan machine (Echosens, France). The CAP measures the attenuation of the US wave on its way through the liver [3]. Similar to the role of the FibroScan in the assessment of LSM, it has been considered as the point-of-care technique for the assessment of HS for a long period of time, due to its availability, non-invasiveness and simplicity of use [15]. Recently, a FAST score (acronym for FibroScan-AST) that unifies a laboratory parameter (AST) with the non-invasive tools (LSM and CAP) was developed to identify patients with high NAFLD activity index scores and advanced fibrosis (F ≥ 2) who are at risk of progressive NASH [16]. 

Recently, a great number of US system manufacturers have developed software for the quantification of HS based on the US physical attenuation phenomenon. 

Some of these methods have already been extensively tested with very good performance reported, while others are still under evaluation [17,18,19]. In the new age, which has brought about an immense amount of non-invasive methods, attenuation imaging has overcome the existing limitations and revolutionized the role of US in the estimation of HS. This manuscript reviews the diagnosis of HS using the conventional B mode US, color Doppler ultrasound (CDUS) and US attenuation imaging methods, including their classification, specifications and performance, with an appraisal on the importance of this armamentarium in daily clinical practice. The PubMed database was systematically searched for relevant literature, concluding on 1 March 2022. We searched using key terms or compounds, appropriate mesh equivalents and their combinations. The following terms were used in the search of the full-text articles: “liver steatosis ultrasound quantification”, “ATI steatosis”, “ATT steatosis”, “CAP steatosis” and “UGAP steatosis”, with 1168 results altogether. The consort flow diagram is systematically presented in Figure 1.

## 2. B-Mode Ultrasound

B-mode ultrasound is a widely available, inexpensive and non-invasive method that precludes exposure to ionizing radiation [2]. Due to these features, it has been endorsed by the European Association for the Study of the Liver (EASL), European Association for the Study of Diabetes (EASD) and the European Association for the Study of Obesity (EASO) as the preferable method for establishing the diagnosis and follow-up of adults with NAFLD [20]. According to the large meta-analysis that included 4720 patients and investigated the accuracy of US compared to LB in diagnosing HS, US has sensitivity and specificity rates of 85% and 94%, respectively, with a 0.93 (0.91–0.95) area under the ROC curve (AUROC) value [21]. However, the ability of US to detect the presence of steatosis was demonstrated only when at least 20% of hepatocytes had become fatty-transformed, making it insensitive for the detection of mild HS.

The main US features for HS include liver brightness (which is increased in comparison to the cortex of right kidney), vessel blurring, posterior US attenuation, impaired diaphragm and gallbladder visualization and focal fatty sparing (FFS) [2].

Vessel blurring manifests as impaired visualization of the borders of the intrahepatic vessels and luminal narrowing, caused by acoustic wave attenuation and vascular remodeling. The increased attenuation of the ultrasonic waves is responsible for the poor visualization of the liver vessels, diaphragm, posterior part of the right liver lobe and the gallbladder wall. FFS represents a zone of the non-steatotic liver parenchyma, without a mass effect on the adjacent vessels or biliary branches, and is usually located in the gallbladder bed; near the falciform ligament; or in periportal, perivenular and subcapsular regions [2,3]. The explanation for the sparing areas are the variances in blood flow leading to the distinct inflow of fatty substrates to hepatocytes. Six different patterns of HS have been described: diffuse, geographic, focal, subcapsular, multifocal and perivascular [22].

Similarly, hyperechoic regions of the focal fatty infiltration (FFI) might also be detected. To distinguish these benign conditions from other focal liver lesions, the application of contrast-enhanced ultrasound (CEUS) is sometimes required. Both FFS and FFI lack the features of other focal but expansive lesions, as they do not compress or displace the nearby vessels nor the biliary ducts.

The morphological and physical features of HS as captured by US have been used in attempts to quantify the severity of steatosis (Table 2) [23].

### Semi-Quantitative Ultrasound Scores

Three semiquantitative scores have been developed to fulfill the need for more precise assessments of HS: the hepatorenal index (HRI), Hamaguchi score and US fatty liver index (US-FLI) [24].

The hepatorenal index is calculated from the brightness ratio between the liver and the renal cortex [25]. The procedure consists of selecting 2 regions of interest (ROI), including the segment VI of the liver and the cortex of the upper pole of the right kidney. The mean brightness of ROI is determined using numerical values assigned to grey-scale pixels, and the index is then calculated [2]. Certain ultrasound devices implement software programs that automatically calculate the index [25,26]. Other authors have separately used software to analyze images in JPEG format or performed histogram analyses on DICOM images [27,28]. Webb et al. used LB as a reference method in a study on 111 patients with chronic liver disease (CLD) and proposed cut-off values for S1, S2 and S3 of 1.49 (AUROC 0.99), 1.86 (AUROC 0.96) and 2.23 (AUROC 0.96), respectively [25]. Petzold et al. defined an HRI cut-off value of 1.46 for ≥S1 with 42.7% sensitivity and 90.7% specificity (AUROC 0.680) on a cohort of 157 patients with CLD [29]. Recently, a retrospective study that included 267 patients who had received an abdominal ultrasound and a subsequent random LB and demonstrated that an HRI ≥ 1.4 corresponds with a >95% positive predictive value (PPV) for HS ≥ 10%, while a HRI ≤ 1.17 has a >95% PPV value for HS ≤ 5% [30]. Stahlschmidt et al. demonstrated that the HRI is unsuitable for estimating HS in patients with advanced liver fibrosis due to the substitution of fat for fibrosis in the evolution of NAFLD. On the group of 34 patients with suspected HS and advanced liver fibrosis, they showed a weak correlation (R = 0.33; *p* = 0.058) of HRI- and MRS-obtained fat fraction scores, with an AUROC of 0.74 when discriminating normal to mild versus moderate to severe steatosis (>15% on MRS) [31]. Similarly, the HRI should not be used in patients with chronic kidney disease, as they might have increased echogenicity of the kidney cortex due to fibrosis accumulation. Although several authors have proven its efficacy in HS quantification, the HRI still lacks validation in large clinical trials [25,27].

In order to improve the evaluation of HS, Hamaguchi introduced the Hamaguchi score in 2007, based on the four US findings: the hepatorenal echo contrast (contrast between the hepatic and right renal parenchyma), liver brightness (high-level echoes arising from the liver parencyhma), deep attenuation (the attenuation of echo penetration into deep parts of the liver and the diaphragm visualization) and vessel blurring (the borders of the liver vessels and luminal narrowing) [32]. According to Hamaguchi et al., HS is defined by a score ≥2 and moderate–severe steatosis by a score ≥4 [32]. Using LB as the reference method, the Hamaguchi score demonstrated sensitivity and specificity rates of 91.7% and 100%, respectively, with a 0.98 AUROC in the detection of HS (≥10%) in 94 Japanese adults with CLD [25]. Another study validated the Hamaguchi score in 167 patients with and without NAFLD for detection of ≥S1, and demonstrated 82% and 100% sensitivity and specificity, respectively, with a 0.94 AUROC in comparison with the CAP as the reference method [33].

Similar to the Hamaguchi score, the US-FLI is based on several sonographic parameters: US beam attenuation; the contrast in echogenicity between the liver and kidneys; the visualization of blood vessels, the diaphragm and the gallbladder wall; as well as the presence of FFS [34,35]. NAFLD is considered confirmed with a score ≥2 [35].

Ballestri et al. evaluated the accuracy of the US-FLI score in 53 patients with HS in ruling out NASH [35]. They concluded that the US-FLI is an independent predictor of NASH (odds ratio (OR) 2.236, *p* = 0.007). A US-FLI <4 had 94% negative predictive value (NPV) in ruling out severe NASH according to Kleiner’s criteria [35]. These results were confirmed in a retrospective study that included 208 patients from the average obese population. The authors used LB as the reference method, and found vessel blurring and poor gallbladder wall visualization to be the most important metrics, but also demonstrated the poor accuracy of the US-FLI in differentiating steatosis from NASH when the US-FLI was ≥5 (AUROC 0.649) [36]. Several years after the initial study, Ballestri defined the US-FLI cut-off values for 3 steatosis grades in the study of 352 biopsied patients with various CLDs, namely mild steatosis ≥2 (AUROC 0.934), moderate ≥3 (AUROC 0.958) and severe ≥5 (AUROC 0.954), with sensitivity rates higher than 86% and specificity rates above 87% [37].

A recently published paper compared the US-FLI with the CAP in 96 patients with NAFLD, and revealed that US-FLI ≥6 had a 94% PPV for steatosis >S2, while US-FLI ≤3 had an NPV of 100% for steatosis >S2, with good discriminative capacity for different steatosis grades (AUROC scores of 0.88 for S1 and 0.90 for S2) [38].

Although simple, these semiquantitive approaches to grade HS have limitations due to the overall insufficient reproducibility of US findings (i.e., low interclass correlation between the readers) and a dependency on the quality of the US equipment. Currently, high-end US machines tend to underestimate the severity of HS due to the technological solutions implemented in order to achieve better US penetration as the response to the requirements from physicians who are facing increasingly overweight patients. In this line of reasoning, B mode images from different manufacturers and classes of US machines might not be comparable in our personal experience.

## 3. Colour Doppler Imaging

When considering the Doppler characteristics of steatotic livers, several authors have demonstrated a monophasic flow in the hepatic veins in a significant proportion of patients, and find it even more specific for characterization of HS than liver cirrhosis [39,40]. Mohammadinia et al. reported the rising incidence of monophasic and biphasic waveforms with the increase in fatty liver severity, with rates of 10%, 55% and 80% for mild, moderate and severe forms of fatty liver disease, respectively [39]. These findings are supported by similar results published by other authors, and are explained by the hepatocyte swelling due to fat infiltration that decreases the liver capsule elasticity and puts pressure on the hepatic veins [40].

According to Alizadeh et al., significant differences were also detected in portal vein (PV) pulsatility, depending on the severity of HS [40]. They found a negative correlation between the PV pulsatility index (PI) and the severity of steatosis, presumably caused by fat infiltration damage to the PV wall. Further hemodynamic disturbances are also seen in the splenic vein, including a decrease in peak systolic velocity (PSV) in association with the HS severity [40].

CDUS has also been verified in the evaluation of FFI and FFS, due to the aberrant venous drainage that often occurs in these areas [41,42]. However, many of these hemodynamic changes have been reported to occur also as the result of fibrosis accumulation in the liver, even in patients with a completely different etiology of CLD, without fatty infiltration, such as viral hepatitis. Therefore, the Doppler findings might be supportive in terms of considering the presence of HS based on B-mode imaging features, but are certainly not specific exclusively for the HS and should not be used as such.

## 4. Quantitative Ultrasound-Based Methods

The goal of quantitative methods is to find a connection between physical properties of the liver tissue (fat vesicles in hepatocytes have different impedance characteristics) and the received US signals dispersed by the tissue. These US signals are then analyzed to estimate the frequency-dependent attenuation and backscatter properties [43].

According to WFUMB, the US techniques used for the quantification of HS may be divided into spectral based techniques, which estimate the attenuation coefficient (AC) and the backscatter coefficient (BSC); and techniques based on the envelope statistics of the backscattered US, which evaluate the acoustic structure quantification (ASQ) and normalized local variance (NLV), or based on the estimation of the speed of sound [3].

The AC measures the loss of US energy passing through the liver, while the BSC measures the US energy returned from the tissue [44].

According to Cook, the use of envelope statistics is a relatively new method, a form of targeted dimension reduction, with the goal of reducing the estimative and predictive variation relative to the standard multivariate statistical methods, sometimes by increasing the sample size many times over [45].

For a better understanding of the subject, the classifications of different US techniques used for the quantification of HS are systematically presented in Figure 2.

### 4.1. Spectral Based Techniques

#### 4.1.1. Controlled Attenuation Parameter (CAP)

The controlled attenuation parameter is a feature added to the FibroScan 502 Touch and 630 systems (Echosens, Paris, France), which is performed using the M (3.5 MHz) or XL probe (2.5 MHz). The CAP measures the attenuation of or reduction in the amplitude of the US waves on their way through the liver as a function of the amount of fat in the hepatocytes, with the results ranging from 100 to 400 dB/m [46,47]. The choice of the most appropriate probe is automatically suggested by the integrated software depending on the skin-to-liver capsule distance (SCD). In cases of SCD >25 mm, the use of XL probe is suggested.


**Procedure**


After fulfilling the general recommendations for participant positioning and breathing (Table 3), the subjects are scanned using the right intercostal approach with the M or XL probe as indicated [48,49]. The FibroScan system estimates both the liver stiffness and CAP using the same US signal, and computes the CAP only when the associated LSM is valid [38]. The specific quality criteria for CAP measurements are not yet clearly defined, although some studies have demonstrated better reliability for CAP measurements when the interquartile range (IQ) over the median CAP value was <30%, or <40 dB/m in terms of absolute values, but this was not confirmed in other studies [46,48,49,50].


**Performance**


There are more than 160 studies that have assessed the accuracy of the CAP in the quantification of HS [51]. Karlas et al. conducted a meta-analysis that included 2735 patients with various liver disease etiologies (hepatitis B, hepatitis C, NAFLD/NASH and others) to evaluate the accuracy of the CAP (measured by M probe only) in HS grading, with histology serving as the reference method. They found a good correlation between the CAP and liver histology, and were able to establish cut-off values of 248, 268 and 280 dB/m for identifying steatosis grades > S0, > S1 and > S2, respectively, with AUROCs of 0.823 for S ≥ 0, 0.865 for S ≥ 1, and 0.882 for S = 3. They found the body mass index (BMI), diabetes and aetiology to have a significant and relevant influence on the CAP values. The authors recommend deducting 10 dB/m from the CAP value for NAFLD/NASH patients and diabetics, because the CAP tended to overestimate steatosis in these groups of patients, probably due to the influence of obesity on the CAP measurements. In line with this, 4.4 dB/m should be deducted for each unit of BMI above 25 kg/m^2^ and added for each unit of BMI exceeding 25 kg/m^2^, independent of the etiology of CLD [52]. Similarly, a study conducted by Ali et al. that investigated the utility of VCTE by FibroScan in the assessment of significant fibrosis (F ≥ 2) in 167 patients with morbid obesity (BMI ≥ 40 kg/m^2^) found higher LSM cut-off values (12.8 kPa, AUROC 0.723, CI 0.62–0.83) as predictors of significant fibrosis [15].

In 2019, Eddowes et al. conducted a multicenter prospective study that included 450 patients with NAFLD for the evaluation of the CAP in comparison to LB. They found acceptable accuracy for the CAP for diagnosing the presence of any liver steatosis with an AUROC of 0.87 (95% confidence interval (CI) 0.82–0.92), whereas its performance for moderate or advanced steatosis was suboptimal (AUROCs of 0.77 (95% CI 0.71–0.82) and 0.70 (95% CI 0.64–0.75), respectively) [50]. The most recent individual patient data meta-analysis included 13 papers with 2346 patients suffering from CLD of various etiologies [53]. The reference method for steatosis quantification was LB, and data acquired using both FibroScan probes (M and XL) were analyzed. This meta-analysis demonstrated that the CAP values were substantially affected by the etiology, presence of diabetes, BMI, aspartate aminotransferase and even sex (on average 13 dB/m higher in males). The optimal cut-offs and their respective AUROCs for detecting any steatosis (S ≥ 0) were 294 db/m (0.807) for NAFLD, 230 dB/m (0.769) for chronic viral hepatitis, 274 dB/m (0.765) for alcoholic liver disease and 244 dB/m (0.687) for other etiologies. Whereas the CAP might be used for steatosis grading among patients with chronic viral hepatitis (AUROC 0.847 for S ≥ 2), it demonstrated suboptimal performance in NAFLD (AUROC 0.736 for S ≥ 2 and AUROC 0.711 for S = 3) [53]. Although the performance of the CAP for detecting any steatosis (S ≥ 1) in NAFLD demonstrated acceptable accuracy (AUROC 0.807), the key question of whether the CAP might be used to screen the individuals under metabolic risk for the presence of any steatosis (which is used to define the presence of NAFLD) remains without a firm answer, given the inappropriate structure of subjects with S0 in studies that evaluated the CAP for NAFLD. The performance of the CAP was also investigated among patients with compensated advanced chronic liver disease, revealing good discriminative ability for detecting any steatosis (AUROC 0.80; 95% CI 0.66–0.90 at 255 dB/m for S ≥ 1) [54].

Several authors have investigated the accuracy of the CAP in comparison to MRI-PDFF, which served as the reference method, and reported moderate to strong correlations between them (r = 0.53–0.809, *p* < 0.001) [47,49,55]. Caussy et al. included 119 adults with and without NAFLD who underwent MRI-PDFF and CAP measurements using either the M or XL probe. The CAP demonstrated good accuracy in detecting initial grades of steatosis as defined by the MRI-PDFF (AUROC scores of 0.80 (95% CI: 0.70–0.90) at the cut-off of 288 dB/m and 0.87 (95% CI: 0.80–0.94) at the cut-off of 306 dB/m for steatosis scores ≥5% and ≥10%, respectively) [49]. Some other studies reported better performance for the CAP in patients with NAFLD for detecting S1 steatosis (AUROCs 0.95–0.97), but insufficient performance to distinguish between moderate and severe steatosis [47,55].

In 180 patients with CLD of various etiologies, using LB as the reference method, Chan et al. demonstrated that the same cut-off values for the CAP may be used for both the M and XL probes in the diagnosis of HS grades [56].


**Ultrasound Attenuation Parameter (UAP)**


In 2013, the FibroTouch FT100 (Wuxi Hisky Medical Technology Co. Ltd., Wuxi, China), which is also a VCTE device, was introduced for determining the UAP in the quantification of HS. The mechanism and the procedure match the FibroScan device.


**Performance**


Ng and coworkers demonstrated a strong correlation (r = 0.76, *p* < 0.001) between the attenuation values obtained using the FibroScan and FibroTouch on the cohort of 163 patients with CLD of various etiologies. They also found the good to excellent intra- and inter-observer agreement [57].

Two histology-controlled UAP studies for determining FibroTouch’s accuracy in defining optimal thresholds of steatosis grades in 816 patients with NAFLD have been conducted [58,59]. The authors found AUROC values of 0.88 (in both studies) for S ≥ 1, 0.77 and 0.93 for S ≥ 2 and 0.70 and 0.88 for S = 3. Still, the optimal UAP cut-off values differed between the studies, with 244 and 295 dB/m for S ≥ 1, 269 and 314 dB/m for S ≥ 2 and 296 and 324 dB/m for S = 3 [58,59].

#### 4.1.2. Attenuation Imaging (ATI)

Attenuation imaging is a two-dimensional (2D) attenuation technique based on B-mode US, which is implemented in the Aplio i-series US machines (Canon Medical Systems, Ōtawara, Japan).

ATI quantifies the degree of the US beam attenuation. In HS, the attenuation increases, especially in deeper areas. When performing ATI, the investigator measures an AC, which represents the loss of US energy passing through the liver, meaning it corresponds to the change in US intensity with depth [60]. The attenuation of the US beam is calculated by analyzing echo signals received by the transducer [61].


**Procedure**


After finding an adequate ultrasonographic window through the intercostal spaces in the right liver lobe, the ATI mode is activated. The B-mode image is displayed on the left side of the screen and the corresponding color-coded ATI sampling box is displayed on the right side [61]. When a color-coded sampling box is positioned, the structures other than the liver parenchyma (such as large blood vessels and biliary ducts) are automatically excluded by the ATI algorithm. A 2 × 4 cm trapezoidal ROI is then placed within the sampling box (Figure 1). While placing the ROI, the investigator should avoid artifacts that manifest as dark orange or dark blue areas. The dark orange areas represent reverberations from the liver capsule and the dark blue artifacts are caused by blood vessels [62]. Sugimoto performed a study comparing three various ROI positions and concluded that placing the ATI-ROI at the upper edge of the sample box should definitely be avoided. The authors recommend that in patients suspected to have severe steatosis, the ATI-ROI should be placed at twice the depth of the liver capsule, while for patients suspected to have mild steatosis the ATI-ROI should be placed at the lower edge of the sample box [62]. The attenuation coefficient (AC) is then measured automatically using the software (dB/cm/MHz). ATI also provides a reliability index (R2) that results in the absence of technical failure of the ATI [8,60]. The R2 values can be classified into three categories: poor (R2 < 0.80), good (0.80 ≤ R2 < 0.90) and excellent (R2 > 0.90). It is considered that an AC with an R2 value ≥ 0.80 indicates a valid measurement [61]. According to the manufacturer’s recommendations, five measurements are performed and the median value is taken as the result [60]. The two important advantages over the CAP are the real time imaging capability with the avoidance of liver vessels or focal liver lesions, as well as the larger sample volume [60]. Usually, a much narrower IQR of ATI values is obtained in comparison to the CAP.


**Performance**


According to the review by Lewis et al. and several subsequently published papers, there are in total 14 studies that have assessed the accuracy of ATI in the quantification of HS compared with the reference methods, as well as their determined cut-offs and AUROC values, which are presented in Table 4 [17,18,19,60,61,63,64,65,66,67,68,69,70,71,72]. Several authors have compared ATI with the CAP, but due to its suboptimal performance in quantifying the liver fat content, the latest WFUMB guidelines recommend against using the CAP as the reference standard [3].

In 2021, Bae compared the performance levels of the B-mode US, MSCT, MRI-PDFF, CAP and ATI methods using the liver histology approach as the reference method in 120 patients who underwent hepatic resection. The MRI-PDFF (AUROC 0.946) outperformed the US (AUROC 0.761), CAP (AUROC 0.829) and MSCT (AUROC, 0.807) methods, while ATI (AUROC, 0.892) proved itself as the second best modality for the detection of any steatosis (≥5% fatty transformed hepatocytes). For moderate steatosis (>33% of fatty transformed hepatocytes), all investigated methods demonstrated good diagnostic performance (AUROCs: MRI-PDFF 0.947, US 0.914, CAP 0.900, MSCT 0.887, ATI 0.914), without significant differences (*p* > 0.05 for all [17]. In 2020, Ferraioli et al. demonstrated only a moderate correlation of the CAP (r = 0.58) with the MRI-PDFF compared to ATI (r = 0.83), with lower AUROCs of the CAP vs. ATI in the detection of HS (AUROC 0.85 vs. 0.92 for S0 vs. S1–S3) [18].

In terms of the reproducibility, high intra-observer (intra-class correlation coefficient (ICC) 0.87–0.96) and inter-observer (ICC 0.79–0.86) agreements were demonstrated when ATI was used for the quantification of HS [73,74].

#### 4.1.3. Attenuation Measurement Function (ATT)

The attenuation measurement function is a software tool developed and available for Hitachi ultrasound machines (Hitachi, Tokyo, Japan, now Fujifilm). With the ATT, two ultrasonic waves of different frequencies are transmitted to the same beamline and the received signal is obtained. Afterwards, the received signal ratio is calculated. The ATT is determined by analyzing the slope on the graph of the computed ratio [75,76,77].


**Procedure**


After placing a patient in the standard position (Table 3), the right intercostal approach is used to mark the ROI and perform the measurement five times in a row. The mean value is taken as the final result [76]. The ATT is measured along the dotted yellow line to the horizontal solid yellow line, which signifies the maximum depth of the ATT measurement (Figure 1) [75,77].


**Performance**


We found one published study that assessed the efficacy of the ATT in comparison with LB [76].

From 2015 to 2017, Tamaki and coworkers conducted a prospective study involving 351 patients with chronic hepatitis, in which they investigated the accuracy of ATT compared to the pathohistological evaluation of liver parenchyma obtained by LB. The obtained cut-off values for the ATT (with AUROC) for grades S1, S2 and S3 were 0.63 (AUROC 0.79), 0.69 (AUROC 0.87) and 0.85 (AUROC 0.96) dB/cm/MHz, respectively. The ATT values increased with an increase in steatosis grade (*p* < 0.001) [76].

Cerit et al. compared unenhanced CT attenuation values of the liver with those obtained via the ATT in 98 adults who underwent a CT evaluation. In detecting mild to moderate steatosis a cut-off value of 0.595 dB/cm/MHz (sensitivity 0.72, specificity 0.73, AUROC 0.830) was detected, while for moderate to severe steatosis an ATT score ≥ 0.665 dB/cm/MHz showed a sensitivity of 100% and a specificity of 90% with an AUROC of 0.935. The ATT proved to have high interobserver agreement (ICC 0.907) [76]. Similarly, Koizumi et al. compared the accuracy of the ATT with the CAP in 94 patients with CLD, and found cut-offs with AUROCs for grades S1, S2 and S3 of 0.68 (AUROC 0.74), 0.72 (AUROC 0.80) and 0.78 (AUROC 0.96) dB/cm/MHz, respectively [78].

#### 4.1.4. Attenuation Plane-Wave Ultrasound (Att.PLUS)

The attenuation plane-wave ultrasound mode measures the decrease in amplitude of ultrasound waves as they propagate throughout the tissue, and is available on the Aixplorer MACH 30 device (SuperSonic Imagine, Aix-en-Provence, France, now Hologic).


**Procedure**


An intercostal approach is used to access the right liver lobe, and afterwards the ROI is placed in a homogeneous area and 3 (according to the manufacturer) to 5 (according to the authors) measurements are performed and expressed in dB/cm/MHz (Figure 1). The final value is the median value, with an IQR/M < 30% [79].


**Performance**


We found only one published study that used the Att.PLUS, in which the CAP was used as the reference method.

Popa et al. performed a monocentric cross-sectional study with 215 NAFLD patients. The authors obtained a cut-off value of 0.5 dB/cm/MHz (AUROC 0.72) for predicting S2–S3 steatosis, and found a moderate correlation (r = 0.45, *p* < 0.001) with the CAP [79].

#### 4.1.5. Ultrasound-Guided Attenuation Parameter (UGAP)

The ultrasound-guided attenuation parameter (UGAP) is the method of steatosis quantification developed by General Electric (GE) Healthcare, which is implemented in the new generation of GE US systems. It is based on a comparison of the measured liver signal and the referential signal (measured on the ultrasound phantom with a known attenuation coefficient) [80]. The basics of this method were first described by Yao et al. in 1990 [81].


**Procedure**


Initially, a single calibration of the ultrasound system is performed using a specific acquisition setup. After placing a patient in the standard position (Table 3), and while using the right intercostal approach to depict the segment V of the liver parenchyma, the activation of the UGAP mode is achieved and a color-coded map is generated. Afterwards, an ROI with a 65 mm length is selected according to standard rules (Table 3), located at least 20 mm from the liver capsule (Figure 1). The UGAP-determined AC is then automatically calculated using the software or the data are stored on a PC and analyzed later. The average of 10 consecutive measurements is considered as the final result [72,80,82].


**Performance**


Six published studies compared the UGAP-determined AC values with the reference methods in the assessment of steatosis grades, in which the authors established cut-off values that seemed similar to those obtained via ATI [72,80,82,83,84,85,86].

Ogino et al. compared the AC obtained using the UGAP with the steatosis grades from liver specimens in 84 patients with NAFLD, and found a significant difference in the AC values among the different grades (*p*< 0.01), as well as an increase in AC values with the steatosis score. The liver fat content determined via a pathohistological evaluation of an LB specimen showed a positive correlation with the AC (r = 0.81, *p*< 0.01), with cut-off values and AUROCs for S1, S2 and S3 of 0.6 (AUROC 0.94), 0.71 (AUROC 0.95) and 0.72 (AUROC 0.88) dB/cm/MHz, respectively [83].

Several other authors also compared the accuracy of the UGAP with LB as the standard in the quantification of HS in patients with CLD, including NAFLD, and also proved a significant correlation between the UGAP-determined AC and steatosis grade, as is systematically presented in Table 4 [72,82,84]. They also compared the efficacy of the UGAP with the CAP, using LB as the refence method, and found the AUROCs of the UGAP in identifying steatosis grades, with significantly better results obtained than with the CAP (0.912 vs 0.842, 0.959 vs 0.817 and 0.924 vs 0.801 for S3) [72,82,84]. Bende et al. found a positive correlation between the UGAP and CAP values in 179 participants with or without CLD (r = 0.73, *p* < 0.0001) [87].

Tada conducted two separate studies including 126 and 608 patients with CLD, and proved the good accuracy of the UGAP in quantifying HS in comparison with the MRI-PDFF (Table 4) [80,85]. The correlation coefficients (*r*) between the MRI-PDFF values and UGAP-obtained AC values were 0.746 (*p* < 0.001) and 0.724 (*p* < 0.001), respectively [80,85]. A large multicentric study conducted in Japan that included 1010 patients with CLD proved the excellent diagnostic accuracy of the UGAP in grading steatosis with reference to the MRI-PDFF. The cut-off values (with AUROCs) for distinguishing steatosis grades ≥1 (MRI-PDFF ≥5.2%), ≥2 (MRI-PDFF ≥11.3%) and 3 (MRI-PDFF ≥17.1%) were 0.65 (AUROC 0.910), 0.71 (AUROC 0.912) and 0.77 (AUROC 0.894) dB/cm/MHz, respectively. They also provided cut-off values for subgroups of patients with NAFLD (≥S1 0.69 (AUROC 0.898), ≥S2 0.72 (AUROC 0.872) and S3 0.79 (AUROC 0.834) dB/cm/MHz) and without NAFLD (≥S1 0.63 (AUROC 0.847), ≥S2 0.67 (AUROC 0.897) and S3 0.77 (AUROC 0.919) dB/cm/MHz) [86].

#### 4.1.6. Tissue Attenuation Imaging (TAI) and Tissue Scatter Distribution Imaging (TSI)

Tissue attenuation imaging and tissue scatter distribution imaging are implemented in the Samsung diagnostic US system (RS80A, Samsung Medison, Co., Ltd., Seoul, Korea).

The tissue attenuation imaging-based AC (AC-TAI) is determined based on the attenuation properties of different frequency components in the tissue, considering that the higher frequency components give higher attenuation rates, contrary to the lower frequency components, and the spectrum of radiofrequency (RF) signals provides a downshift of the center frequency according to depth, where the TAI represents its slope [88].

The issue scatter distribution-imaging based SC (scatter distribution coefficient) (SC-TSI) is a pixel-by-pixel map based on the shape parameter of the Nakagami distribution, reflecting both the concentration and arrangement of the US dispersion [88].


**Procedure**


Considering the general procedure rules (Table 3), the right intercostal plane is used to obtain B-mode images of the liver parencyhma, and the ROI (2 cm inner arc length × 4 cm side length) is placed on TAI and TSI maps below the liver capsule, while avoiding vessels, ducts and focal liver lesions, with a fixed set of time gain compensations and positions of focus. For each parameter, six measurements are obtained and averaged.


**Performance**


One study compared the efficacy of TAI and TSI with the MRI-PDFF as the reference standard on 120 patients with NAFLD [81]. They determined cutoffs for the AC-TAI and SC-TSI of 0.884 db/cm/MHz (AUROC 0.861) and 91.2 (AUROC 0.964) for MRI-PDFF ≥5%, respectively. They also proved a significant correlation of both the AC-TAI and SC-TSI with MRI-PDFF (*r* = 0.659 and 0.727, *p* < 0.001 for both) [88].

Jeon et al. also compared the accuracy of TAI and TSI with the Hamaguchi score and HRI on 243 patients with CLD and healthy volunteers, using CAP as the reference standard. They determined the cut-offs with their AUROCs for both the TAI and TSI parameters of –0.078 MHz/cm (AUROC 0.844) and 0.910 (AUROC 0.827) for ≥S1, respectively. The TAI and TSI parameters provided significantly higher AUROCs for S1 and S2 compared to the Hamaguchi score and HRI (*p* ≤ 0.003). The authors also found a significant correlation of the TSI and TAI parameters with steatosis grades (ρ = 0.593 and ρ = −0.617, *p* < 0.001 for both) determined by CAP [89].

#### 4.1.7. Ultrasound-Derived Fat Fraction (UDFF) and Backscatter Coefficient (BSC)

The ultrasound-derived fat fraction and BSC techniques are implemented in the Siemens Acuson S3000 US system (Siemens Healthineers, Erlangen, Germany).

While the AC measures the loss of ultrasound energy in the tissue, the BSC measures the ultrasound energy returned from the tissue [90]. After the ultrasonic pulses are transmitted into the liver parenchyma, the energy is absorbed and scattered because of the heterogeneous tissue consistency, and a part of the energy is scattered back to the transducer. Afterwards, it is processed and stored using the ultrasound research interface (URI). The BSC measuring unit is 1/cm-sr (the steradian or square radian is the SI unit of the solid angle), and the BSC is considered analogous to the tissue echogenicity [91].

The UDFF is obtained by combining both attenuation and BSC information and is reported as the percentage (%) of HS. In a study of 102 participants, Han et al. developed multivariate QUS models to detect and predict HS. The study used a reference phantom (RP) method, whereby RF data are acquired from the participant and then immediately from an RP without changing the scanner settings [90]. Lately, this technique has been advanced in two ways by Labyed and Mikowski, who introduced the UDFF model—primarily by having RP data integrated into the US system, and additionally by using a fixed-acquisition ROI [43].


**Procedure**


After placing the patient in the standard position, a right intercostal approach is used to access the right liver lobe without major vasculatures, focal lesions or abbreviations.

If measuring the BSC, a 0.6 × 1 cm rectangular ROI is placed on the homogeneous region of the liver parenchyma at least 2 cm below the liver capsule, but not deeper than 7 cm [90].

According to Labyed, when measuring the UDFF, during the participant breath hold phase, acquisitions are performed after placing the 3 × 3 cm ROI 1.5 cm below the liver capsule on a homogenous part of the liver in the fifth or eighth liver segment [43]. The final AC, BSC and UDFF values are then obtained by averaging the estimates from all acquisitions (4–10, depending on the study) [43,92].


**Performance**


We found two published studies that assessed the accuracy of the BSC using the MRI-PDFF as the reference method [90,91].

Lin et al. conducted a prospective, cross-sectional analysis on a cohort of 204 adults with (MRI-PDFF ≥ 5%) and without NAFLD. The primary finding of this study showed that the BSC can accurately diagnose and quantify HS, since they found a good correlation with the MRI-PDFF (ρ = 0.80; *p* < 0.0001). They determined a BSC cut-off of 0.0038 1/cm-sr (AUROC 0.98) for identifying patients with NAFLD (Table 4) [91]. Han et al. found a moderate correlation of the AC (r=0.59, *p* < 0.001) and the BSC (r=0.58, *p* < 0.001) with the MRI-PDFF in a study on 102 participants with known or suspected NAFLD [90]. In another study, Han et al. also demonstrated the good inter-sonographer reproducibility of AC and BSC in adults with known or suspected NAFLD, with ICC scores of 0.86 (95% CI 0.77–0.92) and 0.87 (0.78–0.92), respectively [93].

We found one study that compared the efficacy of the UDFF method using both MRI-PDFF and LB as the reference standards [43]. Labyed and Milkowski confirmed the correlation of UDFF and MRI-PDFF in 101 subjects with known or suspected NAFLD (ρ = 0.87), with a cut-off of 6.34 (AUROC 0.97) for the diagnosis of steatosis (MRI-PDFF ≥ 5%) and a cut-off of 8.1 (AUROC 0.94) for ≥S1 when using LB as the standard [43]. Gao et al. also proved the good intra-observer repeatability and inter-observer reproducibility of UDFF with an ICC *>* 0.85 (95% CI 0.85–0.99) [92].

#### 4.1.8. Liver Fat Quantification

Recenty, Philips released the liver fat quantification (LFQ) tool for the Elite and Affiniti ultrasound systems that measure AC (dB/cm/Mhz).


**Procedure**


After placing the patient and the transducer in the standard position, a rectangular box that presents the confidence map is placed on the homogenous part of the liver parenchyma. It provides a scale of quality from low (0%) to high (100%). A quality score of >60% is recommended by the manufacturer for an accurate measurement. Afterwards, the AC is determined using the round measurement box, which is placed at least 2 cm below the liver capsule, or at greater depth if the confidence map is below 60%.


**Performance**


Up to this point, there have been no published studies that have assessed the accuracy of this ultrasound system in comparison with the reference methods.

One study compared the AC obtained using the Philips Epiq Elite system with the AC obtained using the Canon Medical System Aplio i800 in 30 participants. The CCC (95% CI) was 0.792 and Pearson’s r was 0.839, demonstrating a high agreement between the systems. Both had a mean IQR/M of <15%, although the scores were significantly different between the 2 systems [94].

### 4.2. Techniques Based on the Envelope Statistics of the Backscattered Ultrasound

#### 4.2.1. Acoustic Structure Quantification (ASQ) and Normalized Local Variance (NLV)

The ASQ tool is integrated into the Aplio XG and Aplio 500 (Canon Medical Systems, Tochigi, Japan) as well as the Model 3000 (Terason, Burlington, MA, USA) ultrasound devices. The NLV is implemented in the Aplio i900 system (Canon Medical Systems, Tochigi, Japan).

The ASQ tool measures the focal disturbance (FD) ratio, which is based on the difference between theoretical and real echo amplitude distributions. While the theoretical distribution is based on the probability density function (PDF), also known as the Rayleigh distribution (a continuous probability distribution for positive-valued random variables that represents an ideal homogenous scatter), the real echo amplitude distribution does not appropriately suit this distribution due to structures such as the vessel walls, which are responsible for the heterogenicity in the echo amplitude [95]. This difference between the theoretical and real distribution provides information about the structure of the liver parenchyma. In the case of fatty infiltration, these scatters mask the original small structures, which changes the PDF to more closely resemble a Rayleigh distribution [96]. In other words, the ASQ tool subdivides the primary ROI into a large number of secondary ROIs and displays the histogram from which the mode, average and standard deviation (SD) are derived. The FD ratio is defined as the ratio of the areas under the curve between the real and Rayleigh histograms [97].

The NLV technique is a modification of the ASQ mode by the same manufacturer. They have the same physical characteristics but give different variables. With ASQ, the average value, SD, mode value (value that has higher frequency in a given set of values) and FD ratio are measured. The NLV tool measures the average value and its SD, and does not yield the FD ratio because the NLV is more focused on the homogenicity of the liver, while the FD ratio reflects both the homogenicity and heterogenecity of the liver parenchyma [98].


**Procedure**


After placing the patient in the standard position, the ASQ mode is activated and the measurement is performed five times. Regarding the approach, there are inconsistencies among authors between the epigastric, subcostal and right intercostal approaches, with the last one having a proven correlation with MR spectroscopy (MRS) [99,100]. The focus and depth ranges of the imaging are set at 6–8 and 10–15 cm, respectively [99,101]. The ROIs are marked as large as possible, while avoiding vessels and biliary structures. The FD ratio is then calculated by the software, and the mean ratio of 5 measurements is taken as the final result [101].

The NLV method is performed using the intercostal approach, in the NLV mode, with an ROI diameter of ≥25 mm. The software automatically measures the average value and the SD of the NLV value, and displays the histogram with the NLV value on the *x*-axis and occurrence frequency on the *y*-axis next to the results. The median of five values is taken as the final result.

**Table 4 diagnostics-12-02287-t004:** Studies assessing the accuracy of ultrasound-based methods in the quantification of LS compared to the reference methods (LB and MRI-PDFF) in adults [17,18,19,43,60,61,63,64,65,66,67,68,69,70,71,72,76,80,82,83,84,86,88,91,98].

Authors/Reference		No	Etiology	Reference Method	Method	Cut-Off	AUROC	Cut-Off	AUROC	Cut-Off	AUROC
						S1		S2		S3	
Bae et al.	[53]	108	CLD	LB	**ATI** ** ^1^ **	0.635	0.843	0.7	0.886	0.745	0.926
Tada et al.	[56]	148	CLD	LB	**ATI ^1^**	0.66	0.85	0.67	0.91	0.68	0.91
Jeon et al.	[57]	87	CLD	MRI-PDFF	**ATI ^1^**	0.59	0.76				
Ferraioli et al.	[12]	129	NAFLD and controls	MRI-PDFF	**ATI ^1^**	0.63	0.91	0.72	0.95		
Ferraioli et al.	[11]	72	NAFLD risk	MRI-PDFF	**ATI-PEN ^1^** **ATI-GEN ^1^**	>0.69 >0.62	0.90 0.92				
Dioguardi et al.	[58]	101	CLD	LB	**ATI ^1^**	0.69	0.80	0.72	0.89		
Sugimoto et al.	[59]	111	NAFLD	LB	**ATI ^1^**	0.67	0.88	0.72	0.86	0.86	0.79
Tada et al.	[60]	119	CLD	MRI-PDFF	**ATI ^1^**	0.63	0.81	0.72	0.87	0.75	0.91
Bae et al.	[10]	120	LR for susp. mlg	LB	**ATI ^1^**	0.66	0.914	0.66	0.914		
Lee at al.	[61]	102	NAFLD	LB	**ATI ^1^**	0.64	0.93	0.7	0.9	0.73	0.83
Hsu et al.	[62]	28	CLD	LB	**ATI ^1^**	0.69	0.97	0.78	0.99	0.82	0.97
Kwon et al.	[63]	100	CLD	MRI-PDFF	**ATI ^1^**	0.62	0.91	0.72	0.94		
Jang et al.	[64]	57	LT donors	LB	**ATI ^1^**	0.62	0.808				
Tamaki et al.	[69]	351	CLD	LB	**ATT ^1^**	0.62	0.79	0.67	0.87	0.73	0.96
Fujiwara et al.	[75]	163	CLD	LB	**UGAP ^1^**	0.53	0.9	0.60	0.95	0.65	0.96
Tada et al.	[73]	126	CLD	MRI-PDFF	**UGAP ^1^**	0.60	0.92	0.69	0.87	0.69	0.89
Ogino et al.	[76]	84	NAFLD	LB	**UGAP ^1^**	0.6	0.94	0.71	0.95	0.72	0.88
Kuroda et al.	[77]	202	NAFLD	LB	**UGAP ^1^**	0.49	0.89	0.65	0.91	0.69	0.92
Kuroda et al.	[65]	105	NAFLD	LB	**UGAP ^1^** **ATI ^1^**	0.62 0.64	0.89 0.876	0.72 0.71	0.90 0.88	0.75 0.75	0.91 0.91
Imajo et al.	[80]	1010	CLD	MRI-PDFF	**UGAP ^1^**	0.65	0.910	0.71	0.912	0.77	0.894
Jeon et al.	[81]	120	NAFLD	MRI-PDFF	**TAI ^1^** **TSI**	>0.884 >91.2	0.861 0.964				
Lin et sl.	[84]	204	NAFLD and controls	MRI-PDFF	**BSC ^2^**	0.0038	0.98				
Labyed et al.	[36]	101	NAFLD	LB MRI-PDFF	**UDFF ^3^**	8.1 6.34	0.94 0.97	15.9 /	0.88 /	16.1 /	0.83 /
Bae et al.	[91]	194	CLD or post-OLT	LB	**NLV**	1.095	0.911	1.055	0.974	1.025	0.954
Zhao et al.	[97]	34	MAFLD	LB	**NLV**	1.145	0.875	1.1	0.735	1.1	0.583
Imbault et al.	[100]	17	NAFLD risk	MRI-PDFF LB	**SSE ^4^**	1.541 1.555	0.942 0.952				
Dioguardi et al.	[99]	100	CLD	MRI-PDFF	**SSE ^4^**	≤1.537	0.882	1.511	0.989	1.511	0.989

^1^ Value expressed in dB/cm/MHz. ^2^ Value expressed in 1/cm-sr. ^3^ Value expressed in %. ^4^ Value expressed in mm/μs. Abbreviations: No: number; AUROC: area under the curve; S1: steatosis grade 1; S2: steatosis grade 2; S3: steatosis grade 3; MRI-PDFF: magnetic resonance imaging proton density fat fraction; NAFLD: non-alcoholic fatty liver disease; MAFLD: metabolic-associated fatty liver disease; CLD: chronic liver disease; LB: liver biopsy; LR: liver resection; OLT: orthotopic liver transplantation; susp.mlg: suspected malignancy; ATI: attenuation imaging; ATT: attenuation measurement function; NLV: normalized local variance; SSE: speed of sound estimation; BSC: backscatter coefficient; TAI: tissue attenuation imaging; TSI: tissue scatter distribution imaging; UGAP: ultrasound-guided attenuation parameter.


**Performance**


We have not found any published studies that have compared the accuracy of ASQ with the reference methods, although there are several studies that have compared it with the MRS.

Kuroda et al. and Lee et al. estimated the accuracy of the FD ratio obtained via ASQ when comparing it with a pathohistological assessment of steatosis in mice models. Lee et al. obtained excellent results in terms of diagnostic performance of the FD ratio using 28 male rats, with AUROC values of 1.00 for S1, 0.981 for S2 and 0.965 for S3 [96,102]. The same author compared the diagnostic performance of ASQ with MRS in 36 patients with suspected NAFLD, and found a significant negative linear correlation between the ASQ-derived FD ratio and the HFF (hepatic fat fraction) obtained via MRS in both the initial (ρ= −0.888, *p* < 0.001) and follow-up (ρ= −0.920, *p* < 0.001) examinations [101]. Similar results were reported by Karlas et al. in the cohort of patients with diabetes (n = 50) and healthy controls (n = 20), demonstrating again a negative linear correlation between the FD ratio and HFF (r = −0.43, *p* = 0.004). The authors also found a strong correlation between the ASQ-derived FD ratio and the CAP (r = −0.81, *p* < 0.001) [97]. A negative correlation between the FD ratio obtained via ASQ and the histological grade of HS (r = −0.55; *p* < 0.0001) was also demonstrated in a study that included 51 patients with diffuse hepatopathies [103].

There are two published papers that compared NLV with LB, where the authors obtained cut-offs with their AUROC values [98,104]. We have not found any published study that has compared NLV with the MRI-PDFF in humans.

Bae et al. conducted a prospective study that included 194 participants with various CLD types (n = 97) and patients who underwent liver transplantation (n = 97), and demonstrated the excellent diagnostic performance of NLV as evaluated against the histological grade of HS. The optimal NLV cut-offs for different grades of HS were 1.095 (AUROC 0.911) for ≥S1, 1.055 (AUROC 0.974) for ≥S2 and 1.025 (AUROC 0.954) for ≥S3 [98]. A study with a similar design on a cohort of 34 patients with suspected MAFLD (metabolic-associated fatty liver disease) was published by Zhao et al., who estimated the diagnostic performance and reproducibility of NLV using the 50 mm ROI [104]. They obtained cut-offs with associated AUROCs for mild, moderate and severe steatosis of 1.145 (AUROC 0.875), 1.1 (AUROC 0.735) and 1.1 (AUROC 0.583), respectively. Zhao et al. also demonstrated excellent NLV repeatability (ICC 0.930) [104]. Both mentioned studies showed that the degree of HS is the only significant factor influencing the NLV value in a multivariate analysis (*p* < 0.05 for both) [98,104].

#### 4.2.2. Speed of Sound Estimation (SSE) and Sound Speed Plane-Wave Ultrasound (SSp.PLUS)

In 2017, Imbault and coworkers launched a new method for the quantification of HS from their research institute in Paris, France. It has been named speed of sound estimation and has become available on the Aixplorer ultrasound system (SuperSonic Imagine, Aix-en-Provence, France). The newer version named sound speed plane-wave ultrasound is integrated in the Aixplorer MACH 30 system (Aixplorer, Supersonic Imagine).

This method precisely calculates the speed of sound (SS) in the liver, and is based on the fact that the increase in the fat content decreases the SS [105]. The SS is determined by increasing the spatial coherence of backscattered echoes coming from a targeted focal spot in the medium. First, the distortions of the ultrasound waves caused by the layers of fatty tissue and the muscle layers have to be corrected and accounted for in the SSE calculation. A virtual point-like reflector is created in order to find and correct these aberrations. Afterwards, the correction of the influence of the superficial layer thickness on the SSE is also taken into the calculation [106,107].


**Procedure**


The SSE is performed using the right subcostal window, parallel to the liver capsule, with care taken to avoid large hepatic vessels or artifacts. The acquisition is performed for approximately two seconds and stored in the picture archive. THE SSE is then calculated with appropriate software at a depth of 60 mm. According to THE authors, two to four measurements are enough to calculate the mean value, which is then taken as the final result [106,107]. The muscle and fat thicknesses are measured with conventional US and integrated into the calculation of the final SSE in the liver.

SSp.PLUS is the latest software integrated in the Aixplorer MACH 30 system (Aixplorer, Supersonic Imagine). According to the acquisition protocol proposed by the manufacturer, an intercostal approach is used to access the right liver lobe, the ROI identified in a homogeneous area without large vessels or biliary structures and 3 (according to the manufacturer) to 5 (according to the authors) measurements are performed and expressed in m/s over a range of values from 1450 to 1600 m/s. The final value is the median value, with an IQR/M <30% [108].


**Performance**


Two studies compared the accuracy of SSEs with the MRI-PDFF, and in one of them LB was used as the standard as well [106,107].

In a pilot study, SSEs were performed in 17 patients against the reference methods of the MRI-PDFF and LB. The SSE demonstrated excellent diagnostic performance in the detection of patients with HS, with a cut-off of 1.541 mm/µs (AUROC 0.942) compared to the MRI-PDFF (>5% for HS) and cut-off of 1.555 mm/µs (AUROC 0.952) compared to LB (>10% for HS) [107]. These results were confirmed by Dioguardi-Burgio et al. as well, over a cohort of 100 patients (training group N = 50 and validation group N = 50) who underwent abdominal MR for various indications. According to the MRI-PDFF results, HS was classified as an S1 range of 6.5–16.5%, S2 range of 16.5–22% and S3 values ≥ 22%. At the cut-off ≤ 1.537 mm/µs, the SSE showed good diagnostic ability to detect any grade of HS (AUROC 0.882), and at the cut-off ≤ 1.511 mm/μs an excellent ability to detect moderate–severe HS (AUROC 0.989) in the training group was shown [106].

As for the SSp.PLUS software, it was evaluated in a study that included 133 adult patients with chronic hepatopathies, using CAP as the reference method. The results revealed a strong negative correlation between the SSE and the CAP (r = −0.70, R2 = 0.50, *p* < 0.001). The proposed SSp.PLUS cut-off for predicting the presence of any grade of HS was ≤1537 m/s (AUROC 0.82) [108]. The same group of authors performed an investigation on a cohort of 215 adult patients with NAFLD, and also confirmed a strong correlation between SSp.PLUS and the CAP (r = −0.74, *p* < 0.001). They proposed a cut-off for detection of S2–S3 steatosis (<1524 m/s, AUROC 0.88), and also calculated dual cut-offs optimized for ruling-in steatosis S ≥ 2 (SSp.PLUS ≤ 1516 m/s, specificity 98.36%, sensitivity 58.74%) and for ruling it out (SSp.PLUS ≥ 1559 m/s, specificity 32.8%, sensitivity 95.10%) [79].

## 5. Clinical Significance of Detecting and Grading Liver Steatosis

Due to the high global prevalence of NAFLD, which is currently considered the most common liver disease, there is a paramount need for adequate diagnostic methods for the detection and stratification of patients with fatty liver [109]. Fatty liver is considered the constitutive part of metabolic syndrome, and along with the patient’s age (>50 years), the presence of type 2 diabetes and other metabolic derangements, indicates the risk of the development of CLD and adverse outcomes in relation to this [110,111,112]. These patients should be evaluated for the progressive forms of NAFLD (such as NASH and the presence of advanced fibrosis), as these are related to either faster fibrosis progression (NASH) or overall and liver-related mortality (fibrosis stage) [20]. In this regard, LB remains conditio sine qua non for diagnosing NASH, whereas non-invasive tests including ultrasound-based elastography might be reliably used for fibrosis quantification, according to the guidelines.

With respect to the HS detection and quantification processes, several points are to be considered.

The first refers to the reliability of US-based methods to detect fatty liver, i.e., their performance in distinguishing between the patients with or without HS. Whereas LB and the MRI-PDFF can reliably detect fat accumulation >5%, they are either invasive, expensive or non-available, especially when considering the need for their repetition during patient follow-up. On the other hand, B-mode ultrasound is widely available but has insufficient sensitivity for detecting steatosis <20%. This explains the bursting trends of invention for new non-inasive and relatively inexpensive sonographic methods, which might be used for the prompt identification and quantification of HS. However, many of these methods still need to be tested on large independent cohorts against the reference standards. An additional problem pertaining to their use in NAFLD patients arises from the design of studies that evaluated new devices for HS quantification, since by definition those patients who were included and apparently had S0 steatosis might not be considered as having NAFLD, and this methodological issue has already been highlighted in several papers [53,113].

Another issue is the clinical impact of the steatosis quantification process, given the conflicting results of the studies that have analyzed this problem so far. Whereas previous studies could not demonstrate the association between simple liver steatosis and adverse clinical outcomes, a recent study conducted over the large swedish national cohortrevealed an increased risk of overall mortality even in this group of patients when compared to the matched cohort from the general population [103,114,115]. The risk of all-cause mortality in patients with simple steatosis was increased by 10.7%, in those with NASH without fibrosis was increased by 18.5%, in those with non-cirrhotic fibrosis was increased by 25.6% and in those with cirrhosis was increased by 49.4% [114]. Another recent study found that NAFLD has an even stronger correlation with extrahepatic malignancies than obesity does. In this study, 4722 NAFLD patients were followed for eight years. A total of 2224 incident malignancies were reported, with gastrointestinal cancers accounting for the majority of cases. Thus, strongly supported by these data, simple steatosis might not be viewed as prognostically “benign” as previously believed. According to current theories, NAFLD and NASH cycle back and forth dynamically, with most patients experiencing a slow fibrosis progression [116]. Consequently, the early detection of liver steatosis is essential for providing a well-timed and effective follow-up strategy for preventing liver disease progression.

A reduction in HS can be achieved by applying certain dietary or therapeutic interventions, and this may be monitored via the MRI-PDFF [117]. In this sense, the monitoring of steatosis reductions using ultrasound-based quantitative methods could be a good indicator of therapeutic success, but this should be evaluated in relation to the clinical outcomes. Additionally, the monitoring of steatosis reductions might be an important endpoint in clinical trials of new drugs for NAFLD, for which reliable non-invasive methods for the quantification of HS and liver fibrosis would be desirable diagnostic tools. Given their different technological backgrounds, the quantitative sonographic method that was used for the initial evaluation of HS should also be used during the follow-up with the particular patient, as currently no data exist to demonstrate otherwise.

## 6. Conclusions

When considering which method to choose for the quantification of HS, several characteristics should be taken into account, such as the availability, invasiveness, cost, accuracy and reliability, which have been validated against the reference diagnostic standards and over sufficiently large cohorts of patients.

Besides the CAP, ATI and the UGAP have been the most widely evaluated methods, with promising results. Other methods are under investigation, and have obvious potential due to their clinical utility. According to the published results, it seems that the accuracy of ATI and the UGAP in the assessment of HS could exceed the results achieved with the CAP, but a further assessment requires a head-to head analysis, including other US-based methods, against the reliable reference standards, such as MRI-PDFF or LB. This will be a step forward, along with the development of new drugs for the NAFLD treatment, since the quantification of HS in addition to fibrosis quantification should probably be considered during the initial evaluation, as well as during the follow-up of these patients.

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
