# Peer review of "Ultrasound Methods for the Assessment of Liver Steatosis: A Critical Appraisal"

_diagnostics, 2022, doi:10.3390/diagnostics12102287_

Round 1
Reviewer 1 Report
The review conducted by Bozic et al is much needed, well conducted, and detailed review of the US-based systems available to detect liver steatosis. I congratulate and thank the authors for their contribution. There are some major and other minor concerns that I would like the authors to address. Please see them below.
Ln 70, limitations also include patient discomfort (patients with claustrophobia).
Ln 76, there are many other important ones which includes Dallas Steatosis Index, NAFLD Fibrosis Score, FIB-4, NAFLD liver fat score, FSI, BARD Score, APRI Score, and FAST score. Please discuss these briefly with appropriate references
Ln 91, please add a reference (authors can cite PMID 35268293)
A detailed literature search methodology must be provided along with the article selection/exclusion process. The authors should add a consort flow diagram.
Ln 178-179 sentences are plagiarized, even though the authors cite this study, they must be reworded
Ln 245 font is different
Ln 255, typo (extra period after 4.1.1..)
Ln 274, PMID: 35268293 must be discussed in this session
Ln 324-325, different font
Fig 1. please provide appropriate legend describing each panel.
Ln 358-364 sentences are plagiarized, even though the authors cite this study, they must be reworded
Table 4: Reduce the text size, make it landscape if needed, and improve the readability by keeping the text in horizontal format. Add the appropriate legend. Tables must be stand-alone.
Ln 759-770 different font
Reviewer 2 Report
Even though hepatic fibrosis is the pathological hallmark of the progression of liver disease, the accumulation of excessive hepatic triglyceride, or hepatic steatosis, is increasingly recognized as playing a significant role in the etiology of CLD, and not simply as an "innocent bystander."
The study published by Simon et al. (2021) on the largest available national cohort evaluated the overall and cause-specific mortality in NAFLD patients across time. The authors included over ten thousand middle-aged individuals with histology-confirmed NAFLD investigated between 1966 and 2017 and a control cohort of almost fifty thousand subjects from the general population who were matched for age, gender, calendar year, and county. The key conclusion of this study is that patients with uncomplicated steatosis also showed an elevated risk of all-cause mortality. Patients with simple steatosis had an increased risk of all-cause mortality of 10.7%, those with NASH without fibrosis at 18.5 percent, those with non-cirrhotic fibrosis at 25.6 percent, and those with cirrhosis at 49.4 percent. Another recent study found that NAFLD has an even stronger correlation with extrahepatic malignancies than obesity does. In this study, 4722 NAFLD patients were followed for eight years. A total of 2224 incident malignancies were reported, with gastrointestinal cancers accounting for the majority of case. Thus, strongly supported by these data, simple steatosis is not as prognostically "benign" as previously believed. According to current theories, NAFLD and NASH cycle back and forth dynamically, with most patients experiencing a slow fibrosis progression.
Consequently, early detection of liver steatosis is essential for providing a well-timed and effective follow-up strategy for preventing liver disease progression.
In this article, methods for diagnosing and grading hepatic steatosis that are currently used are reviewed, along with information on how well they perform and how important they are in everyday clinical practice. I believe that the authors have made a complete and easy-to-follow systematization of the available methods for evaluating hepatic steatosis. This review also brings an important update compared to those previously published (2017-2020).
Round 2
Reviewer 1 Report
The authors have well addressed my concerns. No further comments.